# Phylogenomics from transcriptomic "bycatch" clarify the origins and diversity of avian trypanosomes in North America

Spencer C. Galen[1,2,3]*, Janus Borner[2,4], Susan L. Perkins[2,5], Jason D. Weckstein[1,6]

**1** Department of Ornithology, Academy of Natural Sciences of Drexel University, Philadelphia, PA, United States of America, **2** Sackler Institute for Comparative Genomics, American Museum of Natural History, New York, NY, United States of America, **3** Biology Department, University of Scranton, Scranton, PA, United States of America, **4** Institute of Evolutionary Ecology and Conservation Genomics, University of Ulm, Ulm, Germany, **5** Division of Science, The City College of New York, New York, NY, United States of America, **6** Department of Biodiversity, Earth, and Environmental Science, Drexel University, Philadelphia, PA, United States of America

* spgalen@gmail.com

**Data Availability Statement:** All relevant data are deposited in the figshare repository and publicly accessible via the following DOI: 10.6084/m9.figshare.12869966. Additionally, the transcriptome

## Abstract

The eukaryotic blood parasite genus *Trypanosoma* includes several important pathogens of humans and livestock, but has been understudied in wildlife broadly. The trypanosomes that infect birds are in particular need of increased attention, as these parasites are abundant and globally distributed, yet few studies have addressed their evolutionary origins and diversity using modern molecular and analytical approaches. Of specific interest are the deep evolutionary relationships of the avian trypanosomes relative to the trypanosome species that are pathogenic in humans, as well as their species level diversity in regions where they have been understudied such as North America. Here, we address these unresolved areas of study using phylogenomic data for two species of avian trypanosomes that were isolated as "bycatch" from host transcriptome assemblies, as well as a large 18S DNA barcode sequence dataset that includes 143 novel avian *Trypanosoma* 18S sequences from North America. Using a phylogenomic approach, we find that the avian trypanosomes are nested within a clade of primarily mammalian trypanosomes that includes the human pathogen *Trypanosoma cruzi*, and are paraphyletic with respect to the ruminant trypanosome *Trypanosoma theileri*. DNA barcode sequences showed that *T. avium* and an unidentified small, non-striated trypanosome that was morphologically similar to *T. everetti* are each represented by highly abundant and divergent 18S haplotypes in North America. Community-level sampling revealed that additional species-level *Trypanosoma* lineages exist in this region. We compared the newly sequenced DNA barcodes from North America to a global database, and found that avian *Trypanosoma* 18S haplotypes generally exhibited a marked lack of host specificity with at least one *T. avium* haplotype having an intercontinental distribution. This highly abundant *T. avium* haplotype appears to have a remarkably high dispersal ability and cosmopolitan capacity to evade avian host immune defenses, which warrant further study.

data are deposited in the SRA BioProject repository and accessible via the following accession number: PRJNA529266. The sequence data are deposited in the GenBank repository and accessible via the following accession numbers: MT276437-MT276579.

**Funding:** SCG was funded by a Nation Science Foundation Postdoctoral Research Fellowship in Biology (award 1811806) (https://nsf.gov/). JB was funded by the German Academic Exchange Service (DAAD) (https://www.daad.org/en/) with funds from the German Federal Ministry of Education and Research (BMBF) (https://www.bmbf.de/en/index.html) and the European Union (FP7-PEOPLE-2013-COFUND - grant agreement n˚ 605728). The funders had no role in study design, data collection and analysis, decision to publish, or preparation of the manuscript.

**Competing interests:** The authors have declared that no competing interests exist.

## Introduction

Eukaryotic parasites are highly abundant, globally distributed, and of great concern for the health of humans, livestock, and wildlife. However, the majority of eukaryotic parasite biodiversity is poorly known relative to the few species that have become model systems of disease. Fortunately, the development of powerful methods for generating DNA sequence data from non-model parasite groups for which genetic resources are scarce has created a window into the ecology and evolution of many poorly studied parasite lineages. In particular, the bioinformatic extraction of non-target parasite genomic data from host genome and transcriptome assemblies is a promising method for the study of poorly understood parasites and other symbionts [1–5]. Similarly, increasingly large databases of DNA barcodes for eukaryotic parasites from around the world are revealing unexpected levels of parasite genetic diversity and are allowing researchers to estimate species limits in some morphologically cryptic parasite groups [6].

The blood parasite genus *Trypanosoma* (Euglenozoa; Kinetoplastea; Trypanosomatida) is one such group of eukaryotic parasites that is most well-known for parasites that cause disease in humans and livestock such as *Trypanosoma cruzi*, the causative agent of Chagas disease, and *T. brucei*, the causative agent of African trypanosomiasis ("sleeping sickness"). The life-cycles of *Trypanosoma* species vary considerably, but typically consist of one life stage in an invertebrate host that serves as a possible vector for the parasite, and a second life stage in a vertebrate host where the parasite can be found within the bloodstream and other tissues. Transmission of trypanosomes from vector to vertebrate host can occur through a variety of means, including inoculation by the vector, ingestion of the vector by the host, and exposure to vector feces [7–9]. In addition to the well-studied species that cause disease in humans and livestock, the genus *Trypanosoma* also contains a diverse suite of species that are blood parasites of wildlife including mammals, birds, reptiles, amphibians, and fish, and infect a diversity of invertebrates that act as vectors between vertebrate hosts [7, 10–13].

The trypanosomes that infect birds are especially poorly studied, despite being an abundant avian blood parasite throughout the world [12, 14] and having been known to science for over 135 years [15]. In particular, the phylogenetic position of the avian trypanosomes within the broader trypanosome tree of life is controversial and poorly studied. Based on details of their morphology and developmental cycles, the avian trypanosomes were historically hypothesized to be closely aligned with the mammalian trypanosome subgenus *Megatrypanum*, a parasite that infects ruminant livestock [7, 16]. However, this hypothesis was not formally tested until molecular approaches were developed based primarily on sequencing the "barcode" locus 18S rRNA.

A number of molecular phylogenies that include avian trypanosomes have been published since the development of protocols for sequencing 18S, though the topologies have varied considerably. One of the first molecular phylogenetic analyses of the group conducted by Votýpka et al. [17] recovered avian trypanosomes as monophyletic using a small 18S rRNA sequence dataset, though this analysis included just two other species in the genus *Trypanosoma* (*T. cruzi* and *T. mega*, which infect mammals and toads, respectively). Soon after, Votýpka et al. [18] analyzed an expanded 18S rRNA sequence dataset, including 25 non-avian *Trypanosoma* taxa and recovered avian trypanosomes as polyphyletic. This analysis recovered the avian trypanosomes as members of a larger clade primarily containing trypanosomes of squamates, crocodilians, and small mammals (bats and rodents). A more expansive analysis of trypanosomatids recovered avian trypanosomes as sister to a large clade of trypanosomes that infect a broad diversity of vertebrate hosts, though this analysis consisted of only two avian trypanosome isolates that were identified as *Trypanosoma avium* [19]. A subsequent multi-locus

analysis recovered avian trypanosomes as the sister group to a clade containing *T. brucei* and its relatives, though this result was unstable depending on the inference method and data subset analyzed [20].

More recent phylogenetic studies have supported the idea that avian trypanosomes consist of multiple divergent lineages that are not monophyletic. Votýpka et al. [21] analyzed partial gGAPDH sequences and recovered avian trypanosomes as polyphyletic, consisting of one lineage that included *T. avium*, *T. corvi*, and *T. culicavium* that was sister to a clade including *T. brucei*, and an additional distantly related lineage consisting of *T. bennetti*. Zídková et al. [14] conducted the largest analysis of avian trypanosome 18S rRNA sequences to date, including many avian trypanosomes collected from a wide diversity of avian and vector hosts, and found support for three distinct clades of avian trypanosomes (which they called groups A, B, and C). These authors found that collectively the avian trypanosomes were paraphyletic with respect to a diverse clade of mostly mammal trypanosomes, including the causative agent of Chagas disease, *T. cruzi*. Similarly, Šlapeta et al. [22] found three clades of avian trypanosomes (also referred to as groups A, B, and C) using 18S rRNA data. One clade united *T. avium* and a novel taxon, *T. thomasbancrafti*, a second clade consisted of *T. corvi* and *T. culicavium*, while a third clade contained mostly unidentified avian *Trypanosoma* lineages and samples of *T. bennetti*. These authors found that each avian *Trypanosoma* clade was more closely related to trypanosomes that infect other groups of vertebrates than they were to each other. The highly variable position of avian trypanosomes in previously published phylogenies has likely been influenced by the dependence of these analyses on only two molecular markers, 18S rRNA and gGAPDH. No phylogenomic scale analysis of trypanosomes has thus far included samples of avian parasites, rendering our understanding of the evolution of the genus *Trypanosoma* incomplete across deep scales.

Just as avian trypanosome evolution has been understudied across deep time, so has the study of these parasites at the species level. Though nearly 100 species of avian trypanosome have been described [8], molecular data collected over the last 20 years have often conflicted with the original species descriptions that were based on morphology and host associations. Molecular data have revealed that most avian trypanosome species that were described solely on the basis of their avian hosts are likely not valid, as some trypanosome genetic lineages lack the strict host specificity that was assumed by early taxonomic researchers [23]. In addition, molecular data suggest that some described species might actually represent species complexes that contain multiple currently undescribed species. For example, molecular data revealed high genetic diversity and the possible existence of multiple species within the abundant avian trypanosome *T. avium* [14]. One barrier towards understanding avian trypanosome diversity at the species level has been uneven sampling across the world, with North and South America having been particularly undersampled among studies using molecular approaches. The few studies that have reported molecular data for avian trypanosomes from North and South America have tended to focus on relatively few host species within a single geographic area [24, 25], and thus avian trypanosome genetic diversity has likely been undersampled in these regions. The lack of modern genetic sampling from North America is particularly important considering that previous microscopic surveys revealed high abundances of *Trypanosoma* in this region, particularly in the temperate and boreal forests of North America [26, 27]. Lastly, no studies have used statistical approaches such as coalescent species delimitation [28] to identify reproductively isolated lineages of avian trypanosomes, which has the potential to greatly improve our understanding of species-level diversity within this group that is likely characterized by high cryptic diversity [23].

Here, we isolate the first genomic data for avian trypanosomes from the transcriptomes of wild avian hosts and use these data to test phylogenomic hypotheses for the deep evolutionary

history of *Trypanosoma*. We also use a large dataset of avian *Trypanosoma* 18S DNA barcodes to test species delimitation hypotheses for the group within North America. We show that avian trypanosomes are derived from mammal-infecting ancestors and are paraphyletic with respect to ruminant trypanosomes in the subgenus *Megatrypanum*, and that these parasites are likely more diverse within North America than previously realized. By integrating novel data from this study with previously sequenced avian trypanosome isolates from around the world, we show that some haplotypes have strikingly broad host specificities and intercontinental distributions.

## Materials and methods

### Host transcriptome mining for trypanosome sequences

We bioinformatically searched for trypanosome contigs within the avian transcriptomes produced by Galen et al. [4] (SRA BioProject accession: PRJNA529266). These transcriptomes are from 24 avian blood samples (and one liver sample) from North American songbirds that were known to be infected with haemosporidian parasites. Transcriptomic libraries were generated using TruSeq stranded mRNA kits, HiSeq2500 Illumina sequencing (paired-end 125 base-pair reads), and assemblies were produced using Trinity v2.6.6. Complete details of transcriptome generation and assembly for these samples can be found in Galen et al. [4]. We used the ContamFinder pipeline of [1] to identify and filter contigs that were putatively of trypanosome origin. Briefly, this method searches for similarity between contigs of unknown origin from a genome or transcriptome assembly and genomic reference databases for parasites of interest. A detailed description of the ContamFinder method, including code needed to implement this approach, has been previously published [1]. For this study, all available parasite genomes from the Kinetoplastid Genomics Resource TriTypDB (release 42; [29]) were used as reference. Ultimately, the ContamFinder pipeline isolates unambiguous parasite contigs that have a higher similarity to a sequence in the parasite reference database than to any known non-parasite sequence in the UniProt database. As Galen et al. [4] searched for contigs from haemosporidian parasites from the same transcriptomes, we confirmed that no contigs that were bioinformatically assigned to *Trypanosoma* were also found in the set that Galen et al. [4] assigned to haemosporidians.

### Bioinformatic processing

We searched contigs that were identified as being of trypanosome origin for open reading frames (ORFs) of at least 75 amino acids in length using TransDecoder v3.0.1 (http://transdecoder.github.io). We used the resulting ORFs to select the longest isoform per gene identified by Trinity, and eliminated additional isoforms using CD-HIT-EST (-c 0.99, -n 11; [30]). We assigned the remaining contigs to groups of orthologous loci using the OrthoMCL [31, 32] implementation within the EuPathDB database [33]. We selected eight reference genomes from trypanosome species for which complete proteomes are available on TriTrypDB [29] (S1 Appendix) and retained only those orthogroups for which a single ortholog was found in each of the reference species. We produced amino acid alignments for each orthogroup using MAFFT v7.271 [34], implementing the L-INS-I algorithm, and used these alignments to produce codon nucleotide alignments using PAL2NAL [35].

### Phylogenomic analyses

We used codon alignments for maximum likelihood, Bayesian, and species tree phylogenetic analyses. We conducted maximum likelihood analysis using IQ-TREE 1.6.12 [36],

implementing automatic model selection (-MFP), 1,000 ultrafast bootstrap replicates (-bb 1000), and 1,000 SH-like approximate likelihood ratio test replicates (-alrt 1000). We repeated IQ-tree analyses using both concatenated and partitioned by gene datasets. Concatenated datasets were also run using BEAST v2.6 [37], implementing a lognormal relaxed molecular clock (determined by likelihood ratio test against a strict molecular clock model), Yule speciation prior, and 200 million steps (sampling every 20,000 steps with 10% burn-in). The GTR + F + R5 model was used as determined through the IQ-TREE model selection procedure. Maximum likelihood and Bayesian analyses were conducted using both the original alignments and alignments that were run through Gblocks 0.91b [38] to produce a 100% complete data matrix.

Lastly, we estimated a species tree using ASTRAL-III [39]. We generated gene trees for each orthogroup using IQ-TREE with the parameters described above and used the exact version (-x option) to estimate the species tree. We restricted the ASTRAL-III analysis to include only gene trees for which all taxa were represented by sequence data.

## Microscopy

For each sample for which we recovered sufficient transcriptomic data to be included in phylogenomic analyses, we used light microscopy to examine Giemsa-stained blood smears that were made at the time of host sample collection. We examined 50 fields at 200X magnification for trypanosomes and quantified parasitemia as the number of trypanosomes observed in proportion to the number of host red blood cells examined. We then examined each slide at 1,000X to study and photograph *Trypanosoma* trypomastigotes and identify them to morphospecies based on the descriptions in [8, 40, 41].

## 18S rRNA barcoding

To place the avian trypanosomes for which we isolated transcriptomic data into a broader context of previously discovered trypanosome diversity, we amplified and sequenced a fragment of 18S rRNA. We used DNA that was extracted using Qiagen DNeasy kits from blood stored on filter paper at the time of sample collection, and amplified a 770 base pair (bp) fragment of 18S rRNA. We used a nested PCR protocol with outer primer pair Tryp763/Tryp1016 and the nested primer pair Tryp99/Tryp957 [42], using the same cycling protocol for both reactions with an initial step at 95 degrees for 3 minutes followed by 35 cycles of 95 degrees for 45 seconds, 50 degrees for 30 seconds, and 1 minute at 72 degrees. The cycling protocol concluded with a 5 minute step at 72 degrees. All positive amplifications were purified and Sanger sequenced by Functional Biosciences (Madison, Wisconsin, U.S.A.), and sequences were edited using Geneious v8.0.5.

As avian trypanosomes have been poorly studied in North America and there are currently few publicly available sequences for this group from our region of study, we also amplified and sequenced the 18S rRNA fragment for additional trypanosome isolates from Alaska and Pennsylvania, U.S.A. (S2 Appendix). We screened 379 samples in total, including 162 avian blood samples from Alaska and 217 from Pennsylvania. Geographic coordinates for the location where each sample was taken are included in S2 Appendix. Field sampling in Alaska was done under Federal Fish and Wildlife Permit MB779035-1, State of Alaska Department of Fish and Game Scientific permits 16–013 and 17–092, and approved by the Institutional Animal Care and Use Committee of the American Museum of Natural History. Samples from Pennsylvania were collected under Federal Bird Banding permit 23679 and sample collection was approved by the Institutional Animal Care and Use Committee of Drexel University (protocol numbers 20369 and 20689). All amplification and sequencing protocols for these samples were performed as described above.

## 18S rRNA gene tree estimation and species delimitation

We downloaded all avian trypanosome 18S rRNA sequences that were available on GenBank as of 27 December 2019, including isolates of avian trypanosome species that were obtained from vectors. We separately added the sequences from Bernotiene et al. [43] to this dataset. We retained only sequences that covered the same 18S rRNA fragment that we sequenced for this study. We combined the GenBank 18S rRNA data with the novel sequences generated for this study to characterize the distributions and host associations of the haplotypes that we identified in North America within a global context. We used the FaBox DNAcollapser tool [44] to reduce the dataset to unique haplotypes prior to analysis and estimated a Bayesian gene tree using BEAST v2.6, implementing the GTR model, a strict molecular clock, constant-size coalescent tree prior, and 10 million generations (with 10% burn-in). We used *Leishmania amazonensis* isolate (GenBank number JX030088) as the outgroup.

We used a subset of the global 18S rRNA dataset consisting exclusively of samples from North America (including GenBank sequences and novel sequences from this study) to iden-tify reproductively isolated lineages using coalescent species delimitation approaches within the R package *splits* [45]. We implemented the generalized mixed Yule coalescent (GMYC) model [46, 47], which uses a threshold to identify the transition between species-level diver-gences according to the Yule model and intra-species divergences according to a neutral coa-lescent model. We estimated species delimitations using both the single-threshold and multiple-threshold models, which differ in how the model identifies the transition between intra-specific branching and inter-specific branching (the multi-threshold model tests whether additional thresholds improve the fit of the model). The GMYC model requires an ultrametric gene tree as input, and so we used a BEAST gene tree that was generated using the parameters described above and pruned to contain only unique haplotypes, with the outgroup removed as well. We visualized haplotype diversity and relationships among *Trypanosoma* 18S rRNA sequences by constructing a TCS network [48] within PopART [49].

# Results

## Identification of Trypanosoma contigs

We searched for *Trypanosoma* contigs across the 24 avian transcriptomes from Galen et al. [4] and discovered contigs in 18 samples (range: 1 to 1,651 contigs; Table 1). There was no overlap between these *Trypanosoma* contigs and the contigs that were determined to be of haemospori-dian origin from Galen et al. [4]. In the majority of samples we found between 0 and 404 *Trypano-soma* contigs, though in four samples we found considerably higher totals of at least 900 *Trypanosoma* contigs (Table 1). The transcriptome samples from which we obtained at least 900 *Trypanosoma* contigs were from avian hosts of four different families, including an American robin (*Turdus migratorius*) and blackpoll warbler (*Setophaga striata*) from Alaska, U.S.A., and a Baltimore oriole (*Icterus galbula*) and red-eyed vireo (*Vireo olivaceus*) from New York, U.S.A. All four infections that yielded at least 900 *Trypanosoma* contigs had parasitemias of at least 0.002% (range: 0.002–0.03%; Table 1). We tested for whether there was evidence for co-infections with multiple *Trypanosoma* species in any of the transcriptome samples with at least 900 *Trypanosoma* contigs, by searching for multiple contigs from the same sample within a single orthogroup that overlapped and differed at the amino acid level. This test revealed that one of the samples with >900 contigs (SCG369) appeared to be infected with multiple genetically distinct trypanosomes, and so we did not consider this sample further for phylogenetic analyses. All contigs from the 24 transcriptome samples that were identified as being from *Trypanosoma* are available from the fig-share data repository associated with this study (DOI: 10.6084/m9.figshare.12869966).

**Table 1. Results of search for *Trypanosoma* contigs in avian transcriptomes produced by Galen et al. [4].**

| NCBI SRA | Specimen number (Sample number) | Contigs identified by ContamFinder | Contigs remaining after TransDecoder and CD-HIT | Parasitemia (%) |
|---|---|---|---|---|
| SRX5582028 | MSB:Bird:47757 (NK227457) | 302 | 211 | - |
| SRX5582029 | MSB:Bird:47825 (NK277525) | 1 | 0 | - |
| SRX5582027 | MSB:Bird:47846 (NK277546) | 0 | 0 | - |
| SRX5582026 | MSB:Bird:47847 (NK277547) | 305 | 202 | - |
| SRX5582025 | MSB:Bird:48018 (NK277722) | 0 | 0 | - |
| SRX5582024 | MSB:Bird:48026 (NK277730) | 107 | 84 | - |
| SRX5582023 | MSB:Bird:48045 (NK277749) | 2 | 0 | - |
| SRX5582038 | DOT23227 (PRS4407) | 0 | 0 | - |
| SRX5582039 | DOT23232 (PRS4412) | 267 | 191 | - |
| SRX5582040 | DOT23233 (PRS4413) | 353 | 232 | - |
| SRX5582016 | DOT23244 (**PRS4424**) | **1651** | **1134** | **0.012** |
| SRX5582017 | DOT23258 (PRS4438) (blood) | 382 | 261 | - |
| SRX5582018 | DOT23258 (PRS4438) (liver) | 2 | 2 | - |
| SRX5582019 | DOT23268 (PRS4448) | 329 | 221 | - |
| SRX5582020 | DOT23273 (**PRS4453**) | **1539** | **1041** | **0.002** |
| SRX5582021 | DOT23276 (PRS4456) | 5 | 2 | - |
| SRX5582022 | DOT23337 (SCG271) | 404 | 247 | - |
| SRX5582032 | DOT24378 (**SCG296**) | **1357** | **1058** | **0.030** |
| SRX5582033 | DOT24402 (SCG320) | 6 | 4 | - |
| SRX5582030 | DOT24420 (SCG338) | 45 | 34 | - |
| SRX5582031 | DOT24422 (SCG340) | 0 | 0 | - |
| SRX5582036 | DOT24449 (SCG367) | 0 | 0 | - |
| SRX5582037 | DOT24451 (**SCG369**) | **904** | **663** | **0.020** |
| SRX5582034 | DOT24453 (SCG371) | 0 | 0 | - |
| SRX5582035 | DOT24461 (SCG379) | 222 | 156 | - |

Shown are the total number of contigs that were identified by ContamFinder as being from *Trypanosoma*, and the number of contigs that remained following identification of ORFs with TransDecoder and reduction of redundant isoforms using CD-HIT. Four samples were retained for further inspection for co-infections and phylogenomic analysis, in bold: PRS4424, PRS4453, SCG296, and SCG369. The parasitemia values as estimated from microscopy are shown for these samples.

## Microscopic identification of Trypanosoma

We examined blood smears made at the time of sample collection for the three samples with sufficient transcriptomic data for phylogenomic analysis (>900 contigs with no co-infections). Based on the original descriptions and relevant references [8, 40, 41, 50], we identified two morphologically distinct *Trypanosoma* species. We identified the trypanosomes in samples PRS4424 and PRS4453 as *Trypanosoma avium*, based on the elongated body of the trypomastigotes, visible striations, distinctive undulating membrane, and small nucleus relative to the size of the body (Fig 1). The second trypanosome species (sample SCG296) had a compact and short body shape (resembling the shape of a leaf as described by [50]), large nucleus that spanned the width of the body as a band, no visible striations, and posterior position of the kinetoplast (Fig 1). This morphotype most closely resembled the description of *T. everetti*; however, as morphological identification of avian trypanosomes is difficult and we did not have sufficient material to quantitatively assess morphological characters due to low

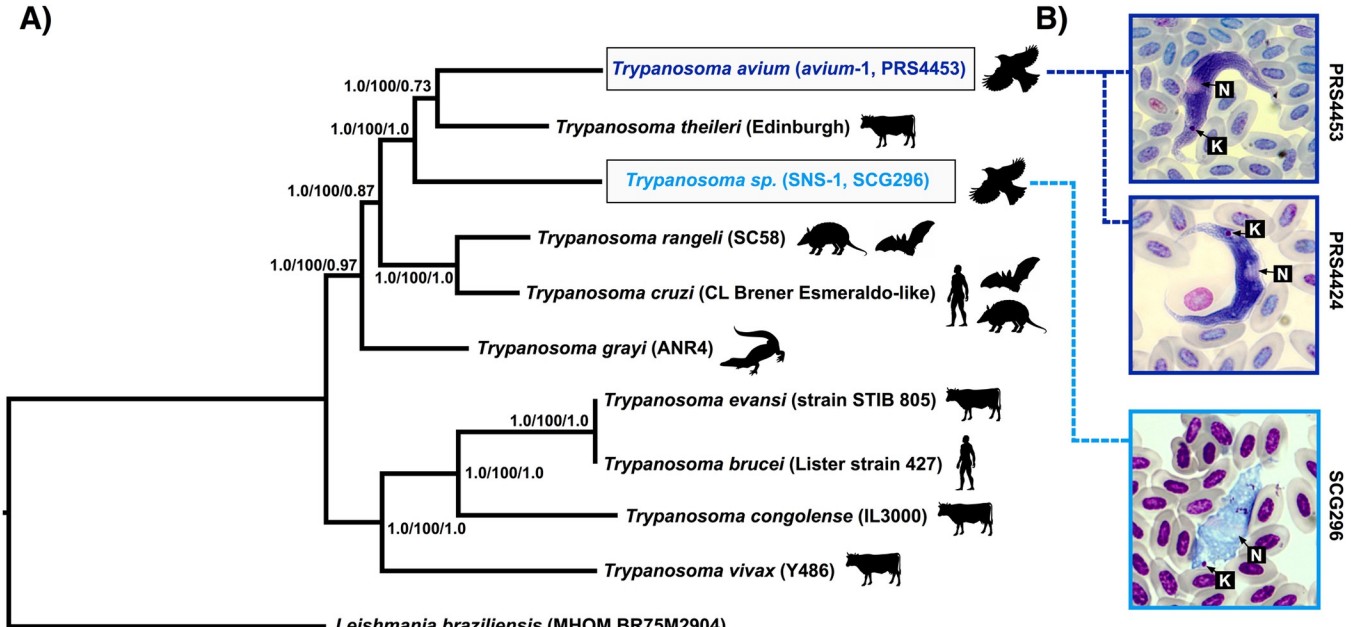

**Fig 1. Phylogenomic hypothesis for *Trypanosoma*.** A) Maximum likelihood phylogenetic hypothesis for the genus *Trypanosoma* based on 321 loci (601,365 base pairs base pairs). Nodal support is shown as Bayesian posterior probability/maximum likelihood bootstrap values/ASTRAL-III local posterior probability. Silhouettes of exemplar host species are shown to the right of each taxon (images from phylopic.org). B) Microscopic images taken at 1,000X of the avian trypanosomes that were included in phylogenomic analyses. Top: *T. avium* sample PRS4453, Middle: *T. avium* sample PRS4424 (not included in phylogenomic analyses but has the same 18S haplotype as PRS4453), Bottom: Sample SCG296 (haplotype SNS-1). Labels on the images are: K = kinetoplast, N = nucleus.

parasitemia, we do not identify this parasite as *T. everetti* at this time. Instead, we refer to this parasite as "*T. everetti*-like" to reflect this uncertainty.

## 18S rRNA barcoding of transcriptome samples

We sequenced a standard 18S rRNA DNA barcode for the three single-infection transcriptome samples that we retained for phylogenomic analysis to put them into a broader context of previously sequenced *Trypanosoma* isolates. We found that trypanosome sequences from the *T. avium* group found in samples PRS4424 and PRS4453 shared an identical 18S haplotype (GenBank accession numbers MT276458 and MT276463) that has been previously detected in multiple host species throughout the world (Table 2). For ease of reference, we refer to this haplotype as "*avium*-1" for the remainder of the study. Trypanosome sequences from sample SCG296 had a divergent 18S haplotype (GenBank accession number MT276498) that differed by 5.8% uncorrected pairwise distance from haplotype *avium*-1. We refer to this haplotype as "SNS-1" (small, non-striated) for the remainder of the study.

## Phylogenomic analyses

As we found that samples PRS4424 and PRS4453 shared the *avium*-1 18S haplotype, we selected the sample that was found to have the lowest proportion of missing data in the final phylogenomic dataset (PRS4453) for further use. We also retained sample SCG296 (haplotype SNS-1), resulting in a final phylogenomic dataset consisting of the eight *Trypanosoma* taxa from TriTryDB, *T. avium* haplotype *avium*-1, haplotype SNS-1, and the outgroup *Leishmania braziliensis*. The transcriptome samples for which we found fewer than 900 total contigs were not used for analysis due to insufficient overlap with the phylogenomic data matrix.

**Table 2. GenBank records with 100% match to haplotypes *avium*-1 and SNS-1.**

| GenBank Accession | Country | Host species | Haplotype |
|---|---|---|---|
| KT728397 | Australia | *Manorina melanocephala* | *avium*-1 |
| KT728398 | Australia | *Xanthomyza phrygia* | *avium*-1 |
| KT728399 | Australia | *Xanthomyza phrygia* | *avium*-1 |
| KT728400 | Australia | *Xanthomyza phrygia* | *avium*-1 |
| KT728401 | Australia | *Xanthomyza phrygia* | *avium*-1 |
| AF416563 | Czech Republic | *Eusimulium securiforme* (vector) | *avium*-1 |
| AY099320 | Czech Republic | *Buteo buteo* | *avium*-1 |
| JN006824 | Czech Republic | *Ornithomyia avicularia* (vector) | *avium*-1 |
| JN006825 | Czech Republic | *Buteo buteo* | *avium*-1 |
| JN006826 | Czech Republic | *Fringilla coelebs* | *avium*-1 |
| JN006829 | Czech Republic | *Accipiter gentilis* | *avium*-1 |
| AB828156 | Japan | *Corvus macrorhynchos* | *avium*-1 |
| AF416559 | Slovakia | *Aquila pomarina* | *avium*-1 |
| MK516191 | Thailand | *Accipiter gularis* | *avium*-1 |
| MK909559 | Thailand | *Gallus gallus* | *avium*-1 |
| MH549546 | Thailand | *Otus sunia* | *avium*-1 |
| FJ649481 | Unknown | 'Buzzard' | *avium*-1 |
| FJ649482 | Unknown | Unknown | *avium*-1 |
| FJ649483 | Unknown | Unknown | *avium*-1 |
| U39578 | Unknown | *Corvus frugilegus* | *avium*-1 |
| FJ649480 | Unknown | Unknown | *avium*-1 |
| KX179915 | U.S.A | *Icteria virens* | SNS-1 |

Ultimately, we only retained orthogroups that included sequences from at least one of the avian trypanosome transcriptome samples, resulting in a full phylogenomic dataset consisting of 321 loci (601,365 base pairs). Sample PRS4453 *avium*-1 was represented at 175 loci and sample SCG296 SNS-1 was represented at 197 loci, though both samples were characterized by high total proportions of missing data with sample PRS4453 *avium*-1 having 84.3% missing data and SCG296 having 86.1% missing data. As a result, we also analyzed a dataset that used Gblocks to produce a 100% complete matrix (9,545 base pairs total). All alignments used for analyses are provided in the figshare data repository associated with this study (DOI: 10.6084/m9.figshare.12869966).

All analyses (maximum likelihood, Bayesian, and species tree) recovered two deeply divergent clades (Fig 1), regardless of partitioning scheme or use of the Gblocks subset with 100% matrix completion. The first clade consisted of *T. brucei*, *T. evansi*, *T. congolense*, and *T. vivax*, with all analyses recovering the same topology of *T. brucei* sister to *T. evansi* and *T. vivax* as the sister lineage to the rest of the clade. The second deep clade consisted of *T. grayi*, *T. cruzi*, *T. rangeli*, *T. theileri*, and the two avian trypanosomes (PRS4453 *avium*-1 and SCG296 SNS-1). All analyses recovered two subclades, with one containing *T. cruzi* and *T. rangeli*, and the second containing *T. theileri* and the two avian trypanosomes. *T. grayi* was consistently found as the sister lineage to these two subclades. Within the subclade containing the avian trypanosomes, the avian trypanosomes were always found as paraphyletic with respect to *T. theileri* (Fig 1).

## 18S sequence diversity and species delimitation

From GenBank we downloaded 232 avian *Trypanosoma* 18S sequences that we were able to align, of which 166 overlapped mostly or entirely with the fragment of 18S that we sequenced.

We generated novel *Trypanosoma* sequence data for 143 isolates from North America, including 42 from boreal forest sites in Alaska, U.S.A., and 101 from a temperate forest site in Pennsylvania, U.S.A. (GenBank accession numbers: MT276437-MT276579, S2 Appendix). The novel 18S isolates revealed that the *avium*-1 and SNS-1 haplotypes occur at high frequencies in North American bird communities, with 38 of 42 isolates from Alaska belonging to *avium*-1 and 76 of 101 isolates from Pennsylvania belonging to SNS-1.

We used the total 309 sequence dataset (166 from GenBank and 143 novel sequences) to characterize global distributions and host specificities of *Trypanosoma* haplotypes, while we used a restricted dataset consisting of 147 avian *Trypanosoma* sequences from North America for species delimitation analyses. The FaBox DNAcollapser tool [44] reduced the global dataset to 84 haplotypes and the North America dataset to 17 haplotypes, which we used to build Bayesian gene trees. The global 18S gene tree revealed that haplotypes with morphological identifications generally clustered together, but were typically not monophyletic with respect to haplotypes without morphological identifications (Fig 2, S1 Fig). Haplotype *avium*-1 clustered with several other haplotypes that have been attributed to *T. avium* in a larger clade that contained *T. thomasbancrofti*. Haplotype SNS-1 fell into a large clade that contained haplotypes of *T. anguiformis* and *T. bennetti*, though it was not phylogenetically close to haplotypes that have been morphologically identified as *T. everetti* from Bernotiene et al. [43]. The global (309 sequence), North America (147 sequence), and GMYC (17 sequence) 18S alignments are available from the figshare data repository associated with this study (DOI: 10.6084/m9. figshare.12869966).

The gene tree containing only North American haplotypes was input for the GMYC model. Both single-threshold and multiple-threshold GMYC models were not significantly different from a null model in which all sequences represent a single species. However, as our dataset included haplotypes from the *T. avium* and SNS-1 groups which are morphologically, genetically, and ecologically distinct, we view the hypothesis that all haplotypes are from a single species to be unlikely. Accordingly, we report the maximum likelihood estimates for the number of entities from the GMYC models below, as we believe they provide a useful approximation of avian *Trypanosoma* lineage diversity in North America.

The maximum likelihood number of entities in our sample ranged between three (single-threshold model; confidence interval 1–16) to nine (multiple-threshold model; confidence interval 1–9). The single-threshold model identified one lineage that consisted of haplotypes primarily from Pennsylvania that contained haplotype SNS-1 as well as two isolates sampled in Missouri, U.S.A. from Soares et al. [25]. The single-threshold model also identified a lineage that consisted of haplotypes primarily from Alaska that included haplotype *avium*-1, and a third lineage that contained three haplotypes that were sampled at three different sites (Pennsylvania, Missouri, and Alaska; Fig 3). Similarly, the maximum likelihood estimate of the multiple-threshold model also identified a lineage that consisted of haplotype SNS-1 and other haplotypes from eastern North America, though this model split eight additional haplotypes into distinct entities (Fig 3). Out of the 17 haplotypes included in this analysis, we were able to obtain microscopic images of the trypomastigote morphologies of seven haplotypes (S2 Fig). These images show consistent differences in qualitative morphological characters (primarily overall size and the presence/absence of striations) among the three primary clades in the North America avian *Trypanosoma* gene tree.

## Discussion

Avian *Trypanosoma* have been known to science for well over a century, though our knowledge of the relationships of these parasites to other trypanosomes as well as their species-level

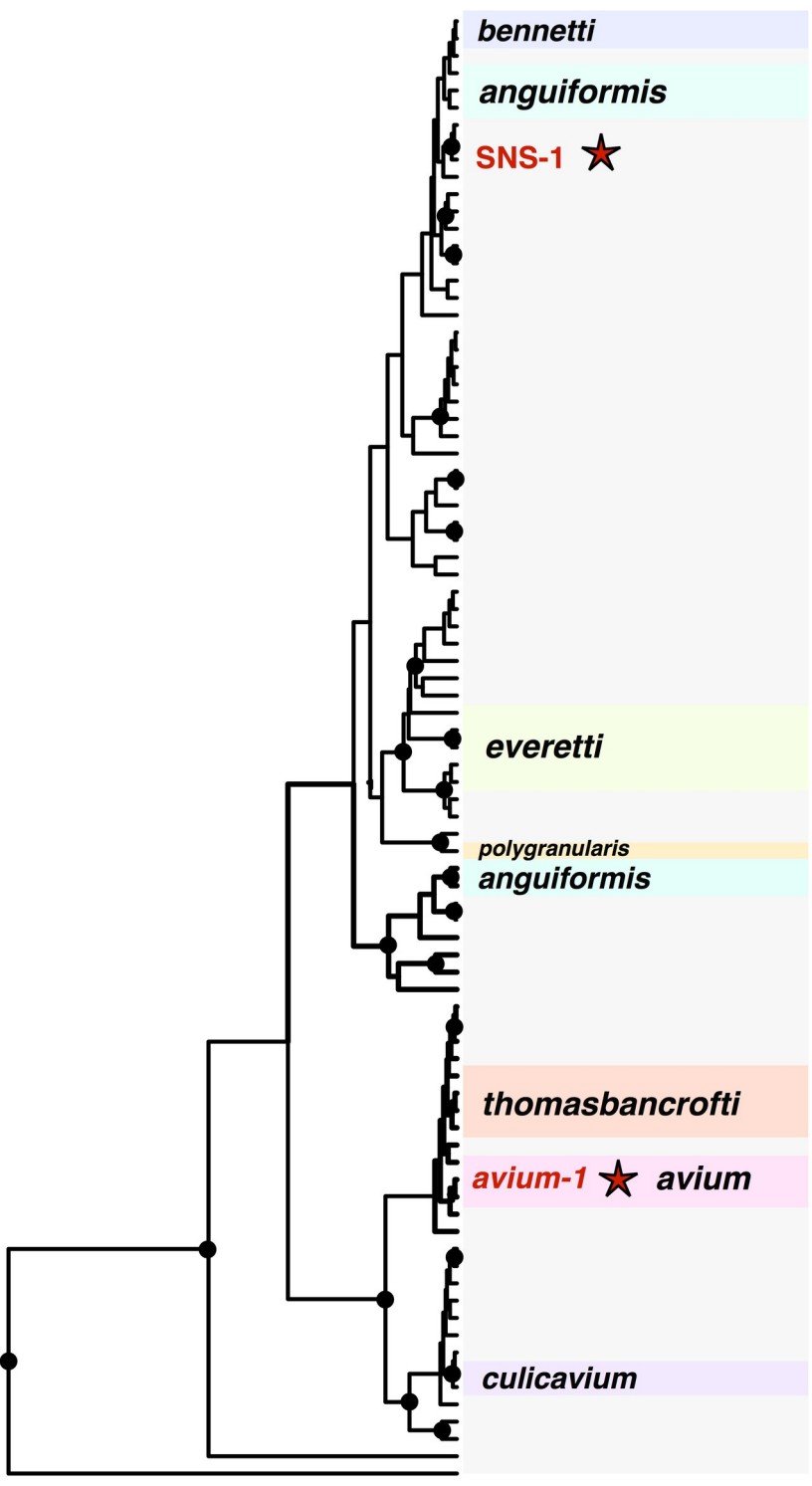

**Fig 2. Global avian *Trypanosoma* 18S gene tree.** BEAST gene tree for 84 avian *Trypanosoma* haplotypes isolated in this study or obtained from GenBank. Groups of haplotypes that have been morphologically identified according to their GenBank entry are labeled by species. The two 18S haplotypes that we found to be abundant in North America, *avium*-1 and SNS-1, are labeled in red and marked by stars.

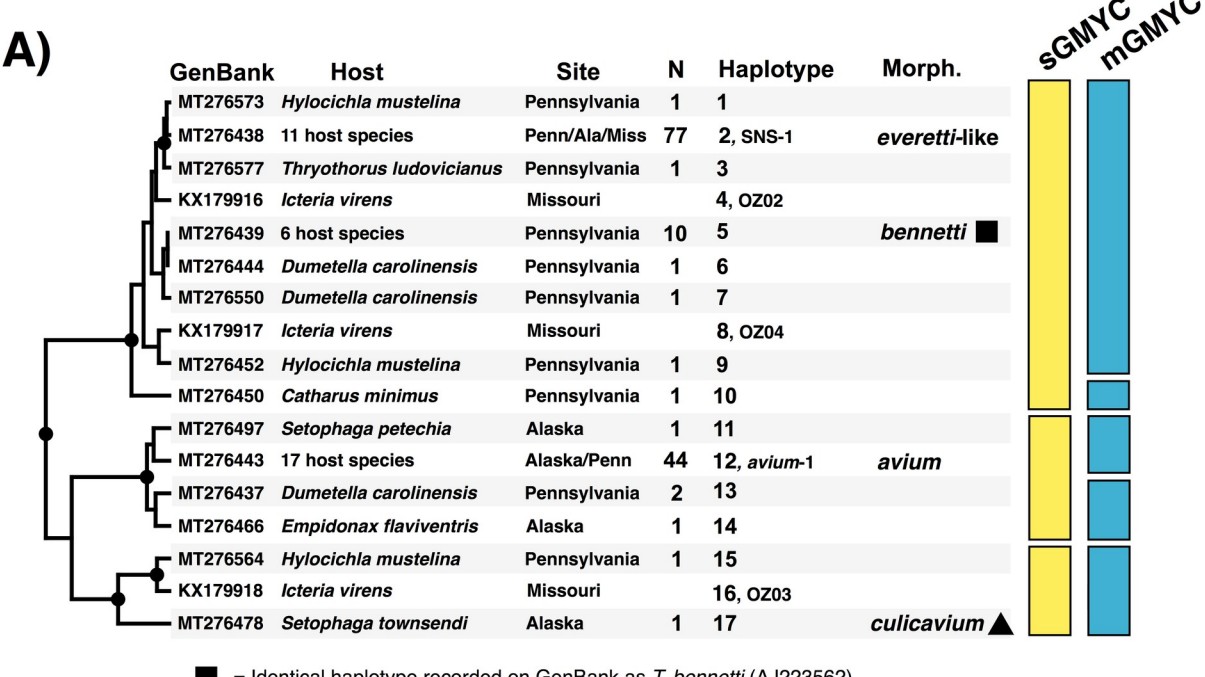

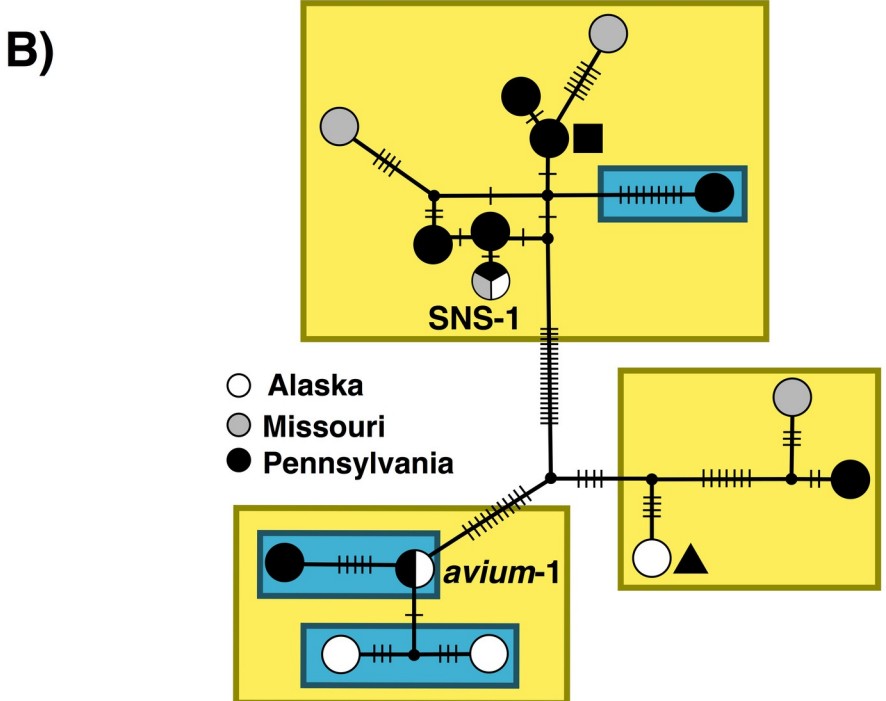

**Fig 3. 18S gene tree and species delimitation results for North American avian trypanosomes.** A) Shown to the right of the phylogeny are known host species, sampling sites in North America, number of times each haplotype was sampled, and haplotype name (either given in this study or in previous studies). Column 'sGMYC' indicates the results of the single-threshold GMYC model from the 'splits' R package, with contiguous blocks indicating haplotypes that were identified as distinct entities in the maximum likelihood model. Column 'mGMYC' shows the equivalent for the multiple-threshold GMYC model. B) TCS haplotype network for North American avian trypanosomes with species delimitations depicted (yellow boxes indicate lineages that were supported in the sGMYC and the mGMYC analysis, smaller green boxes indicate lineages that were supported only in the mGMYC analysis).

diversity have remained understudied in the modern era of advanced molecular and analytical techniques. By bioinformatically searching avian blood transcriptomes for parasite 'bycatch' and extracting *Trypanosoma* contigs from infected samples, we present here the first phylogenomic hypothesis for the relationships of avian trypanosome species within the broader phylogeny of *Trypanosoma*. We also conducted the most comprehensive analysis of avian *Trypanosoma* diversity in North America that has been conducted to date, incorporating all publicly available 18S rRNA barcode sequences and generating 143 novel sequences. Our approach integrating phylogenomic and DNA barcode data emphasizes the need for an increased focus on avian trypanosomes, as it is impossible to understand the evolutionary history of the globally important genus *Trypanosoma* without considering the diversity of these bird-infecting parasites.

## Avian trypanosomes are closely related to the mammal trypanosome subgenus *Megatrypanum*

Avian trypanosomes have been represented in a number of previous phylogenetic hypotheses for *Trypanosoma*, though the analysis presented here is the first to include them in a phylogenomic scale analysis. Previous analyses have used 18S rRNA or other single-locus approaches and have recovered the avian trypanosomes in varied positions largely depending on variation in taxon sampling. Here, we show with strong support that avian trypanosomes in two distinct morphospecies groups (*T. avium* and an unidentified morphospecies similar to *T. everetti*) are nested within the clade that is classically referred to as Stercoraria. In this analysis Stercoraria was represented by *T. cruzi*, *T. rangeli*, *T. theileri*, and *T. grayi*. All analyses recovered a close relationship between the avian trypanosomes and the ruminant trypanosome *T. theileri*, with *T. theileri* nested within the avian trypanosomes as sister to the *T. avium* group. This result is noteworthy for several reasons. First, this relationship supports an early hypothesis for the evolutionary history of the trypanosomes that postulated a close relationship between avian trypanosomes and the subgenus *Megatrypanum* (of which *T. theileri* is a member) based on similarities in morphology and developmental cycle [7, 16]. Specifically, several studies have now found that avian trypanosomes develop in the hindgut of their vectors and are likely transmitted by ingestion of the vector or by mechanical means through the introduction of vector feces into abraded skin [9, 51, 52]. This transmission route is similar to the mechanical transmission of *Megatrypanum* [7], and contrasts with the transmission of other trypanosomes like those in the subgenus *Salivaria*, which are characterized by parasite introduction directly into the host through the bite of the vector when feeding [7].

The close evolutionary relationship between avian and mammalian trypanosomes is also noteworthy as it supports a role for host-switching in driving *Trypanosoma* lineage diversification. As has been supported in earlier analyses for this clade, we found strong support for the non-monophyly of mammalian trypanosomes [20]. In all analyses we found that the sister group to the predominantly mammalian 'Stercoraria clade' is the crocodilian trypanosome *T. grayi*, with the avian trypanosomes nested within the group of mammalian trypanosomes that includes *T. cruzi*, *T. rangeli*, and *T. theileri*. This topology suggests that, similar to other groups of blood parasites such as the haemosporidians [52, 53], host switching between mammalian and sauropsid vertebrate hosts has occurred frequently and has driven the diversification of trypanosomes. Future phylogenomic studies that include a broader diversity of *Trypanosoma* lineages, including representatives of the "aquatic clade" of trypanosomes that infect aquatic vertebrates [54] and the trypanosomes that infect squamates, will provide greater resolution to how host-switching has impacted the evolutionary history of the trypanosomes. In particular, the inclusion of other avian trypanosome species will be essential to address whether there

have been two independent transitions to infecting birds, as previous analyses using small sequence datasets have found some support for the polyphyly of avian trypanosomes [21].

## The diversity of avian trypanosomes in North America

The study of avian trypanosomes has not only lagged behind research on the globally important trypanosomes of mammals, but also other groups of avian blood parasites such as the malaria parasites and other haemosporidians [6]. The lack of recent attention to the avian trypanosomes is somewhat surprising considering the high abundance of these parasites in avian host communities [12], though several factors such as the typically low infection intensities of avian trypanosomes and the difficulty of morphologically identifying previously described species [42] have likely contributed to their current status as an understudied group. However, the ubiquity of avian trypanosomes in bird communities throughout the world [8] and their potential for virulence [55] suggest that increased attention is warranted to further our understanding of host-parasite evolutionary ecology.

Of particular need are surveys of the trypanosome communities of currently understudied regions of the world using both molecular and microscopic approaches. In an effort to place the samples for which we obtained transcriptome data within a broader geographic and species-level context, we conducted the first multi-community molecular survey of avian trypanosomes in North America and generated novel 18S rRNA DNA barcode sequence data for 143 *Trypanosoma* isolates from 379 screened avian hosts. This survey revealed that the two samples that we included in our phylogenomic analyses reflect high-abundance *Trypanosoma* 18S rRNA haplotypes that are common in North American bird communities. The 18S haplotype that was found in the transcriptomes of samples PRS4424 (host *Turdus migratorius*) and PRS4453 (host *Setophaga striata*), which we refer to as haplotype *avium*-1, was found at high frequency in our boreal forest sample sites in Alaska (90% of isolates from Alaska belonged to this haplotype). The haplotype that was found in transcriptome sample SCG296 (host *Vireo olivaceus*), which we refer to as haplotype SNS-1, was similarly abundant in the temperate forest communities that we sampled in Pennsylvania (75% of sequenced isolates from temperate forest belonged to this haplotype).

Based on a comparison of *avium*-1 and SNS-1 to all previously sequenced avian *Trypanosoma* isolates that are available on GenBank, it appears likely that haplotype SNS-1 is restricted to North America as it has only been previously recorded from the host *Icteria virens* in Missouri, U.S.A. [25]. In contrast, haplotype *avium*-1 appears to have an exceptionally broad geographic distribution, having been previously found in samples taken in Asia, Europe, Australia, and at both the boreal forest and temperate forest sites in North America that we sampled for this study (Table 2). Intercontinental distributions as observed for *avium*-1 are emerging as a common phenomenon among avian trypanosome species. For example, Šlapeta et al. [22] described the avian trypanosome *T. thomasbancrafti* from Australian honeyeaters, but noted that this taxon likely has a distribution that includes Europe based on previous records from GenBank. Similarly, Cooper et al. [56] noted that four species of avian *Trypanosoma* from Australia likely have intercontinental distributions. It remains to be seen following more in depth surveys whether exceptionally broad geographic distributions are typical of avian trypanosome species, or whether restricted distributions such as the apparent limitation of haplotype SNS-1 to North America are the norm.

Haplotypes *avium*-1 and SNS-1 also exhibited a remarkable lack of host specificity: within our newly sampled isolates, we found *avium*-1 in 17 host species (7 host families), while SNS-1 infected 10 host species (8 host families). After including host records from GenBank, the observed host breadth for each haplotype was even greater– 28 host species (12 host families)

for *avium*-1, and 11 host species (9 host families) for SNS-1 (S1 Table). These patterns of host specificity conform to previous inferences from both microscopic [41, 51, 52] and molecular [14, 23, 56] studies showing a marked lack of vertebrate host specificity for avian trypanosomes. The extreme host generalism that is exhibited by avian trypanosomes is striking in contrast to other avian eukaryotic blood parasites such as haemosporidians, which are often highly specialized to specific host lineages [57, 58]. Interestingly, the host generalism of avian trypanosomes contrasts with the host specificity that is observed in some mammalian trypanosomes [59]. The molecular mechanisms that underlie the ability of avian trypanosomes to infect such a broad phylogenetic diversity of hosts is currently unknown and deserves increased study.

We used statistical species delimitation approaches to test whether the *avium*-1 and SNS-1 haplotypes are reproductively isolated lineages and estimate whether there are other reproductively isolated avian trypanosome lineages in North America. Surprisingly, both GMYC models that we tested were not significantly different from a null model of a single species. We believe that this finding is likely attributable to the low variability of the 18S locus [42, 60] and resulting lack of power of the model to detect between-species divergences, rather than the existence of just a single avian trypanosome species in North America. The observation that haplotypes *avium*-1 and SNS-1 are dramatically different in morphology, are nearly 6% divergent across 18S, and exhibit distinct distributions strongly suggests that they are reproductively isolated species, and indicates that the GMYC model was underpowered to identify species divergences using our 18S dataset. Haplotype *avium*-1 appeared to represent the morphological species *T. avium*, which was supported by the observation that identical 18S haplotypes are identified as this species on GenBank. In contrast, the species identity of haplotype SNS-1 was unclear. Though this haplotype qualitatively resembled the species *T. everetti*, a global 18S gene tree revealed that SNS-1 is distantly related to isolates that have been morphologically confirmed to be *T. everetti* by Bernotiene et al. [43]. The uncertainty surrounding the species identity of SNS-1 speaks to the importance of increasing global sampling of avian *Trypanosoma* isolates in the future, particularly those that are linked to microscopic images for morphological analysis.

Though neither GMYC model was significantly different from a null model, each model identified clusters of sequences that are worthy of discussion. Most importantly, the maximum likelihood estimates of both species delimitation models (single-threshold GMYC and multiple-threshold GMYC) recovered 18S haplotypes *avium*-1 and SNS-1 as distinct entities. This finding supports the general use of morphological characters for species delimitation of *Trypanosoma* [42], as these two haplotypes reflected distinct morphologies. This result is not surprising considering the high pairwise sequence distance between *avium*-1 and SNS-1 (5.8%), as morphologically distinct avian trypanosomes have been described with 18S sequence divergence under 1% [60]. In addition to *avium*-1 and SNS-1, the GMYC models identified several other distinct entities of avian *Trypanosoma* in North America that may represent reproductively isolated lineages. Interestingly, both threshold models identified a divergent clade as a distinct entity that contained haplotypes from across North America (Fig 3), including a haplotype that was a 100% match to GenBank sequences that were identified as *T. culicavium* from Europe. The haplotype that matched to *T. culicavium* was divergent from both *avium*-1 (3.5%) and SNS-1 (5.1%), which is within the range of divergence of avian *Trypanosoma* species that have been described on the basis of morphology [60]. This finding suggests that at the very least there is a third species of avian trypanosome in boreal/temperate forests of North America, if not more. Interestingly, a haplotype (GenBank accession MT276439) that was supported by both species delimitation analyses as part of the same evolutionary lineage as SNS-1 has been previously identified as the morphological species *T. bennetti* (GenBank accession: AJ223562). This observation further supports that the species delimitation estimates shown

here are highly conservative, and that the North American avian trypanosome fauna is likely far more diverse than currently understood.

In conclusion, this study represents an important framework for future analyses of avian *Trypanosoma* phylogenetic relationships and diversity at local and global scales. We present the first phylogenomic hypothesis for the genus *Trypanosoma* that includes avian trypanosomes, confirming a close evolutionary relationship between this group and the mammal trypanosomes of the subgenus *Megatrypanum*. We also placed the avian trypanosomes from the phylogenomic analysis into a broader regional context of avian *Trypanosoma* diversity in North America. We found that two divergent 18S haplotypes are the dominant lineages of trypanosomes in North American boreal and temperate forest bird communities, respectively, but that there may be additional unrecognized species-level lineages of avian trypanosome that deserve further study. Several directions that we have initiated in this study should continue to be pursued, including linking morphological variation of avian *Trypanosoma* to DNA barcode and genomic datasets, the use of multi-locus DNA sequence datasets to investigate *Trypanosoma* evolutionary relationships and species limits, and surveys of *Trypanosoma* diversity in regions of the world that have not been recently studied using molecular approaches.

## Supporting information

**S1 Fig. BEAST gene tree of global avian *Trypanosoma* 18S haplotypes.** Haplotypes *avium*-1 and SNS-1 are highlighted in red and marked with a star.
(TIFF)

**S2 Fig. Microscopic images of North American avian trypanosomes.** Shown are microscopic images that correspond to the trypanosome 18S haplotypes in the gene tree to the left. Isolate sample numbers are shown to the right of each image.
(TIFF)

**S1 Table. Host species for 18S haplotypes *avium*-1 and SNS-1.** Documented host species for *Trypanosoma* 18S haplotypes *avium*-1 and SNS-1 from this study and previously sequenced isolates on GenBank.
(DOCX)

**S1 Appendix. *Trypanosoma* and *Leishmania* isolates from the TriTrypDB database that were used for identification of orthogroups and phylogenomic analyses.**
(XLSX)

**S2 Appendix. Host and sampling site information for novel *Trypanosoma* 18S rRNA sequences from this study.**
(XLSX)

## Acknowledgments

We thank Gediminas Valkiūnas for advice regarding avian trypanosome research and Janice Dispoto for assistance in the lab. We thank Lisa Kiziuk and the Willistown Conservation Trust for permission to sample birds at the Rushton Woods Preserve, and Alison Fetterman, Blake Goll, Todd Alleger, Emily Ostrow, Suravi Ray, Kevin Fitzpatrick, and Moed Gerveni for assistance in the field.

## Author Contributions

**Conceptualization:** Spencer C. Galen.

**Data curation:** Spencer C. Galen.

**Formal analysis:** Spencer C. Galen, Janus Borner.

**Funding acquisition:** Spencer C. Galen.

**Resources:** Susan L. Perkins, Jason D. Weckstein.

**Writing – original draft:** Spencer C. Galen.

**Writing – review & editing:** Spencer C. Galen, Janus Borner, Susan L. Perkins, Jason D. Weckstein.

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
