## [Decision Letter · Decision Letter 0]

22 Jul 2020

PONE-D-20-16707

Phylogenomics from transcriptomic “bycatch” clarify the origins and diversity of avian trypanosomes in North America

PLOS ONE

Dear Dr. Galen,

Thank you for submitting your manuscript to PLOS ONE. After careful consideration, we feel that it has merit but does not fully meet PLOS ONE’s publication criteria as it currently stands. Therefore, we invite you to submit a revised version of the manuscript that addresses the points raised during the review process.

We look forward to receiving your revised manuscript.

Kind regards,

Tzen-Yuh Chiang

Academic Editor

PLOS ONE

Journal Requirements:

2. We note that you are reporting an analysis of a microarray, next-generation sequencing, or deep sequencing data set. PLOS requires that authors comply with field-specific standards for preparation, recording, and deposition of data in repositories appropriate to their field. Please upload these data to a stable, public repository (such as ArrayExpress, Gene Expression Omnibus (GEO), DNA Data Bank of Japan (DDBJ), NCBI GenBank, NCBI Sequence Read Archive, or EMBL Nucleotide Sequence Database (ENA)). In your revised cover letter, please provide the relevant accession numbers that may be used to access these data. For a full list of recommended repositories, see http://journals.plos.org/plosone/s/data-availability#loc-omics or http://journals.plos.org/plosone/s/data-availability#loc-sequencing.

3. In your Methods section, please provide additional location information of the collection sites, including geographic coordinates for the data set if available.

Reviewers' comments:

Reviewer's Responses to Questions

**Comments to the Author**

1. Is the manuscript technically sound, and do the data support the conclusions?

Reviewer #1: Yes

Reviewer #2: Partly

2. Has the statistical analysis been performed appropriately and rigorously? 

Reviewer #1: Yes

Reviewer #2: N/A

3. Have the authors made all data underlying the findings in their manuscript fully available?

Reviewer #1: Yes

Reviewer #2: Yes

4. Is the manuscript presented in an intelligible fashion and written in standard English?

Reviewer #1: Yes

Reviewer #2: Yes

5. Review Comments to the Author

Reviewer #1: Galen et al. have produced a high-quality piece of work which helps clarify the evolutionary history of avian trypanosomes and contributes a significant amount of data and new knowledge to the field of parasite phylogenomic. Their use of transciptome mining is commendable and a technique that is growing in use as the amount of transcriptomic data from non-model organisms and wildlife populations grows. I would encourage the authors to publish in more detail their code/analysis pipeline in order to facilitate these methodologies in the future; although I acknowledge that bioinformatics development is beyond the scope of the current work under review.

Overall I do not have any concern regarding their methods and I concur that their agree with their conclusions and that they are drawn accurately from the data and analyses presented. Please ensure that all figshare supporting documentation is fully available upon publication with active doi’s. There are several (optional) grammatical suggestions I have for the introduction (see below), however overall the manuscript is written and presented to a high quality.

Line 59: “eg” not needed in reference.

Line 61: “…is one such group of eukaryotic parasites that is most well-known for parasites that cause disease in humans and livestock…”

Line 64: “The lifecycles of Trypanosoma spp. vary considerably, …”

Line 69-70: “… well-studies species that cause disease in humans and livestock…”

Line 71: “… and fish, and infect a diversity of invertebrates that act as vectors between vertebrate hosts.”

Line 73-75: repeated from above, not needed again.

Reviewer #2: I like this MS combing two different aspects focusing on avian trypanosomes. By using a phylogenomic approach (which represents a novelty tool for avian trypanosomes), the authors find that avian trypanosomes are nested within a clade of primarily mammalian trypanosomes (and suggest that avian trypanosomes are derived from mammal ancestors and are paraphyletic concerning the subgenus Megatrypanum). This finding corresponds (and verify) the previous studies on this topic. In addition to this, they also conducted the most comprehensive analysis of avian Trypanosoma diversity in North America generating 143 novel sequences. 18S rRNA barcode sequence analysis demonstrates that avian trypanosomes are likely more diverse within North America than previously realized. The authors also found that avian Trypanosoma 18S haplotypes generally exhibit a marked lack of host specificity (e.g., T. avium haplotype 1). This was also demonstrated previously for some avian trypanosome species; however, the authors’ statement could not be applied generally for all avian trypanosomes – more likely this is a mix of host-specific (restricted) haplotypes as well as generalists. Thus the abstract statement “These findings suggest that avian trypanosomes have the potential for remarkably high dispersal abilities and cosmopolitan capacities to evade avian host immune defenses, which warrant further study.” is rather overestimated, even though some genotypes/haplotypes could have such a capacity. I believe that the results of the study are sufficiently interesting and that there is, therefore, no need for a similar exaggeration in the interpretation of the results obtained.

I have just two comments:

I am not sure if the name T. everetti is used correctly. How can be the authors sure that it is T. everetti, which was described from Africa? According to the phylogenetic analysis, their sequences belong to the T. bennetti clade. How did they distinguish their everetti-1 from T. bennetti? To be honest, I am afraid, that the use of the name T. everetti is more likely premature and will complicate further study on this topic. Did the authors compare their everetti-1 with the previously described bennetti-morphotypes? Did they compare morphospecies from slides on which the genospecies are identical to T. bennetti (e.g., MT276439) with morphospecies everetti-1? Such a comparison would verify if these different haplo/genotypes differ also in their morphology. I do not understand why morphotypes of all different haplotypes are not presented in the study (at least as supplementary material; this should be added).

My second comment focuses on the diversity of avian trypanosomes in North America. It is not clear why only North-America haplotypes are analyzed – such an analysis does not show if some haplotypes are shared between continents and do not demonstrate the position of newly detected haplotypes within the other known species/genotypes, etc… This must be analyzed and demonstrated at least as supplementary material.

(The main three monophyletic groups (Fig. 2 A) should be matched to the known (morpho)species, as T. bennetti, T. avium, and T. culicavium. I understand, that this is presented as a note, but names in the tree would be more illustrative.)

60-1 Euglenozoa; Kinetoplastea; Trypanosomatida

65 a possible vector

101-15 also study of Slapeta et al. (2016) and his analysis should be mentioned in this paragraph.

501-2 Table 2 does not include the mentioned information!!! Tab. 2 includes only previously reported sequences identical with two mentioned haplotypes!!! This information must be presented in a more clear form.

Fig. 2A: MT276439 – 6 host species, but there is no way to find which six species belong to this haplotype, according to the supp. material MT276439=Icterus galbula, but other five species? The supplementary material should be better arranged.

PLS correct T. culicavum to T. culicavium throughout the text (and Fig. 2A)

6. PLOS authors have the option to publish the peer review history of their article (what does this mean?). If published, this will include your full peer review and any attached files.

Reviewer #1: No

Reviewer #2: No

---

## [Author Response · Author response to Decision Letter 0]

3 Sep 2020

Dear Dr. Chiang,

We thank you and the two reviewers for your constructive and thoughtful review of our manuscript. We were able to address all comments made by both reviewers, and we believe you will find the manuscript improved and suitable for publication. In response to Reviewer #1, we have added more detail regarding our bioinformatic approach and have modified several sections to make our writing clearer. We have also followed the recommendation of Reviewer #2 and in the revised manuscript we no longer identify the species Trypanosoma everetti, as this reviewer suggested that more caution was warranted. We agree with the reviewer’s suggestion, and in the revised manuscript refer to this parasite only by its 18S haplotype identity to reduce confusion regarding its species-level identity. Reviewer #2 also suggested several additional figures and tables, which we now provide as a new figure in the main text (Figure 2 in the revised text), and in the Supplement (S1 Table, S1 Figure, and S2 Figure). In sum, we believe that you will find the revised manuscript improved in clarity and in the presentation of our results after having addressed all reviewer comments.

Sincerely, 

Spencer Galen (on behalf of all authors)

Response to Journal Requirements:

###

### Author’s response:

###

We have ensured that our revised manuscript meets PLOS ONE’s style requirements.

2. We note that you are reporting an analysis of a microarray, next-generation sequencing, or deep sequencing data set. PLOS requires that authors comply with field-specific standards for preparation, recording, and deposition of data in repositories appropriate to their field. Please upload these data to a stable, public repository (such as ArrayExpress, Gene Expression Omnibus (GEO), DNA Data Bank of Japan (DDBJ), NCBI GenBank, NCBI Sequence Read Archive, or EMBL Nucleotide Sequence Database (ENA)). In your revised cover letter, please provide the relevant accession numbers that may be used to access these data. For a full list of recommended repositories, see http://journals.plos.org/plosone/s/data-availability#loc-omics or http://journals.plos.org/plosone/s/data-availability#loc-sequencing.

###

### Author’s response:

###

We have deposited all data from this study in appropriate databases, and accession numbers are listed in the revised manuscript:

Transcriptome data: SRA BioProject accession: PRJNA529266 (line 157)

18S sequences: GenBank accession numbers: MT276437-MT276579 (line 374)

3. In your Methods section, please provide additional location information of the collection sites, including geographic coordinates for the data set if available.

###

### Author’s response:

###

We have noted in the revised manuscript that geographic coordinates for each sample are available in the S2 appendix. Because the samples for this study were taken from a large number of distinct sampling locations, it is not prudent to list them individually in the body of the manuscript. Lines 230-232:

“We screened 379 samples in total, including 162 avian blood samples from Alaska and 217 from Pennsylvania. Geographic coordinates for the location where each sample was taken are included in S2 Appendix.”

###

### Author’s response:

###

We have now deposited all original data files from this study in a figshare data repository. The DOI for this repository is: 10.6084/m9.figshare.12869966

Response to Reviewer’s comments:

Reviewer #1: Galen et al. have produced a high-quality piece of work which helps clarify the evolutionary history of avian trypanosomes and contributes a significant amount of data and new knowledge to the field of parasite phylogenomic. Their use of transciptome mining is commendable and a technique that is growing in use as the amount of transcriptomic data from non-model organisms and wildlife populations grows. I would encourage the authors to publish in more detail their code/analysis pipeline in order to facilitate these methodologies in the future; although I acknowledge that bioinformatics development is beyond the scope of the current work under review.

###

### Author’s response:

###

We thank the reviewer for their comments regarding the quality of our work. To improve the description of our bioinformatic methods as requested by this reviewer, in the revised manuscript we have slightly expanded this section of the manuscript to include more detail and make it easier for other researchers to find previously published works from our group that describe these methods in a more robust manner. We specifically note that two previously published papers describe in great detail the same analysis pipeline that is used in this study: Galen et al. 2020, Molecular Ecology Resources was focused on the broader analysis pipeline that is implemented here, and Borner et al. 2018, BMC Genomics was focused specifically on the ContamFinder pipeline that was integral to this study. To avoid repeating this information and making the manuscript under review unnecessarily long, we refer readers to these works. Lines 156-174 of the revised manuscript:

“We bioinformatically searched for trypanosome contigs within the avian transcriptomes produced by Galen et al. [4] (SRA BioProject accession: PRJNA529266). These transcriptomes are from 24 avian blood samples (and one liver sample) from North American songbirds that were known to be infected with haemosporidian parasites. Transcriptomic libraries were generated using TruSeq stranded mRNA kits, HiSeq2500 Illumina sequencing (paired-end 125 base-pair reads), and assemblies were produced using Trinity v2.6.6. Complete details of transcriptome generation and assembly for these samples can be found in Galen et al. [4]. We used the ContamFinder pipeline of [1] to identify and filter contigs that were putatively of trypanosome origin. Briefly, this method searches for similarity between contigs of unknown origin from a genome or transcriptome assembly and genomic reference databases for parasites of interest. A detailed description of the ContamFinder method, including code needed to implement this approach, has been previously published [1]. For this study, all available parasite genomes from the Kinetoplastid Genomics Resource TriTypDB (release 42; [29]) were used as reference. Ultimately, the ContamFinder pipeline isolates unambiguous parasite contigs that have a higher similarity to a sequence in the parasite reference database than to any known non-parasite sequence in the UniProt database. As Galen et al. [4] searched for contigs from haemosporidian parasites from the same transcriptomes, we confirmed that no contigs that were bioinformatically assigned to Trypanosoma were also found in the set that Galen et al. [4] assigned to haemosporidians.”

Overall I do not have any concern regarding their methods and I concur that their agree with their conclusions and that they are drawn accurately from the data and analyses presented. Please ensure that all figshare supporting documentation is fully available upon publication with active doi’s. 

###

### Author’s response:

###

We have now deposited all data relevant to this study in a figshare repository (DOI: 10.6084/m9.figshare.12869966).

There are several (optional) grammatical suggestions I have for the introduction (see below), however overall the manuscript is written and presented to a high quality.

Line 59: “eg” not needed in reference.

###

### Author’s response:

###

We have made this change in the revised manuscript (line 61).

Line 61: “…is one such group of eukaryotic parasites that is most well-known for parasites that cause disease in humans and livestock…”

###

### Author’s response:

###

We have made this change in the revised manuscript (lines 62-64).

Line 64: “The lifecycles of Trypanosoma spp. vary considerably, …”

###

### Author’s response:

###

We have made this change in the revised manuscript (lines 65-66).

Line 69-70: “… well-studies species that cause disease in humans and livestock…”

###

### Author’s response:

###

We have made this change in the revised manuscript (line 71).

Line 71: “… and fish, and infect a diversity of invertebrates that act as vectors between vertebrate hosts.”

###

### Author’s response:

###

We have made this change in the revised manuscript (lines 73-74).

Line 73-75: repeated from above, not needed again.

###

### Author’s response:

###

We have removed this sentence in the revised manuscript.

Reviewer #2: I like this MS combing two different aspects focusing on avian trypanosomes. By using a phylogenomic approach (which represents a novelty tool for avian trypanosomes), the authors find that avian trypanosomes are nested within a clade of primarily mammalian trypanosomes (and suggest that avian trypanosomes are derived from mammal ancestors and are paraphyletic concerning the subgenus Megatrypanum). This finding corresponds (and verify) the previous studies on this topic. In addition to this, they also conducted the most comprehensive analysis of avian Trypanosoma diversity in North America generating 143 novel sequences. 18S rRNA barcode sequence analysis demonstrates that avian trypanosomes are likely more diverse within North America than previously realized. 

###

### Author’s response:

###

We thank the reviewer for their positive comments regarding our study.

The authors also found that avian Trypanosoma 18S haplotypes generally exhibit a marked lack of host specificity (e.g., T. avium haplotype 1). This was also demonstrated previously for some avian trypanosome species; however, the authors’ statement could not be applied generally for all avian trypanosomes – more likely this is a mix of host-specific (restricted) haplotypes as well as generalists. Thus the abstract statement “These findings suggest that avian trypanosomes have the potential for remarkably high dispersal abilities and cosmopolitan capacities to evade avian host immune defenses, which warrant further study.” is rather overestimated, even though some genotypes/haplotypes could have such a capacity. I believe that the results of the study are sufficiently interesting and that there is, therefore, no need for a similar exaggeration in the interpretation of the results obtained.

###

### Author’s response:

###

We agree with the reviewer that our statement in the abstract about Trypanosoma host specificity was overstated. In the revised manuscript we have made this statement more specific to reflect that evidence for high dispersal abilities and low host specificity is currently only available for the abundant avium-1 haplotype. Lines 45-47 in the revised manuscript:

“This highly abundant T. avium haplotype appears to have a remarkably high dispersal ability and cosmopolitan capacity to evade avian host immune defenses, which warrant further study.”

I have just two comments:

I am not sure if the name T. everetti is used correctly. How can be the authors sure that it is T. everetti, which was described from Africa? According to the phylogenetic analysis, their sequences belong to the T. bennetti clade. How did they distinguish their everetti-1 from T. bennetti? To be honest, I am afraid, that the use of the name T. everetti is more likely premature and will complicate further study on this topic. Did the authors compare their everetti-1 with the previously described bennetti-morphotypes? Did they compare morphospecies from slides on which the genospecies are identical to T. bennetti (e.g., MT276439) with morphospecies everetti-1? Such a comparison would verify if these different haplo/genotypes differ also in their morphology. 

###

### Author’s response:

###

We thank the reviewer for their detailed comments regarding avian Trypanosoma taxonomy. After some contemplation, we agree with the reviewer that it is in the best interest of the field not to identify the parasite in question as T. everetti at this time. However, we believe that T. everetti remains the most likely identity of haplotype “everetti-1” and the other similar haplotypes that we identified in this study. Though the reviewer notes that T. everetti was described from Africa, we would like to point out that it has since been found in multiple studies from North America and South America, suggesting that this species is widespread across this region. Specifically, T. everetti has been identified in Mexico (Bennett et al. 1991), Costa Rica (Valkiūnas et al. 2004), the United States (Arizona; Deviche et al. 2005), and Colombia (Matta et al. 2004). 

The reviewer also notes the possibility that our everetti-1 haplotype actually belongs to T. bennetti, because of the close phylogenetic relationship between haplotype everetti-1 and a sequence from GenBank that was identified as T. bennetti. While this remains a possibility, it is not possible to determine if this is the case because the original description of T. bennetti is lacking in detail of the morphology of the trypomastigote form with no associated images with which to compare (Kirkpatrick and Terway-Thompson 1985). We also note that the phylogenetic association of everetti-1 and the GenBank sequence attributed to T. bennetti is not strong evidence of shared morphology, as previous research has shown that avian trypanosomes can be close phylogenetically yet have dramatically different morphologies (Sehgal et al. 2015).

As we recognize the importance of using caution when working with groups with challenging taxonomies, in the revised manuscript we refer to the species in question as unidentified and have replaced all uses of “T. everetti” to “T. everetti-like” or “resembling T. everetti”. In addition, we no longer refer to the associated 18S haplotype as “everetti-1”, instead calling it “SNS-1” to emphasize two of its morphological characters (small size and non-striated appearance). We believe this modification should make it clear that while the morphology of this parasite resembles the description of T. everetti, we cannot rule out other possibilities without detailed morphological study (which was beyond the scope of this manuscript). We have also added text to explain our use of the “T. everetti-like” terminology. Lines 308-315 of the revised manuscript:

“The second trypanosome species (sample SCG296) had a compact and short body shape (resembling the shape of a leaf as described by [50]), large nucleus that spanned the width of the body as a band, no visible striations, and posterior position of the kinetoplast (Fig 1). This morphotype most closely resembled the description of T. everetti; however, as morphological identification of avian trypanosomes is difficult and we did not have sufficient material to quantitatively assess morphological characters due to low parasitemia, we do not identify this parasite as T. everetti at this time. Instead, we refer to this parasite as “T. everetti-like” to reflect this uncertainty.”

References:

Bennett, G.F. et al. 1991. Blood parasites of some birds from northeastern Mexico.

Deviche, P. et al. 2005. Interspecific differences in hematozoan infection in Sonoran desert Aimophila sparrows.

Kirkpatrick, C.E. and Terway-Thompson, C.A. 1985. Biochemical characterization of some raptor trypanosomes. II. Enzyme studies, with a description of Trypanosoma bennetti n. sp.

Matta, N.E.M. et al. 2004. Prevalence of blood parasites in tyrannidae (flycatchers) in the eastern plains of Colombia. 

Sehgal, R.N.M. et al. 2015. Trypanosoma naviformis sp. nov. (Kinetoplastidae: Trypanosomatidae) from widespread African songbirds, the Olive sunbird (Cyanomitra olivacea) and Yellow-whikered greenbul (Andropadus latirostris).

Valkiūnas, G. et al. 2004. Additional observations on blood parasites of birds in Costa Rica. 

I do not understand why morphotypes of all different haplotypes are not presented in the study (at least as supplementary material; this should be added).

###

### Author’s response:

###

We have added microscopic images of the morphology for seven 18S haplotypes in Figure 3 (which was called Figure 2 in the original version of the manuscript) as a supplementary file (Figure S2). Unfortunately, we were unable to assess the morphology of all haplotypes due to restricted access to these samples caused by the COVID-19 pandemic (which are currently in a lab in New York City that none of the authors currently have permission to enter). Although morphological data for all haplotypes would be a valuable contribution to this field of research, we note that this was not a primary focus of our study. We believe that the microscopic images that we were able to obtain will, however, be an important resource moving forward for the study of avian trypanosomes.

My second comment focuses on the diversity of avian trypanosomes in North America. It is not clear why only North-America haplotypes are analyzed – such an analysis does not show if some haplotypes are shared between continents and do not demonstrate the position of newly detected haplotypes within the other known species/genotypes, etc… This must be analyzed and demonstrated at least as supplementary material.

###

### Author’s response:

###

We agree with the reviewer’s suggestion that adding a global analysis would be a valuable addition to our study. In the revised version of the manuscript, we have added a global phylogenetic analysis of 84 avian Trypanosoma 18S haplotypes, including novel haplotypes generated by this study as well as haplotypes that were obtained from GenBank. We were also able to add haplotypes of T. everetti from a newly published study by Bernotiene et al., which provided important insights for our analysis. This new analysis resulted in a new Figure 2 as well as a new supplementary Figure (S1 Fig). This analysis and its results are discussed in the following lines of the revised manuscript:

Methods (lines 242-252):

“We downloaded all avian trypanosome 18S rRNA sequences that were available on GenBank as of 27 December 2019, including isolates of avian trypanosome species that were obtained from vectors. We separately added the sequences from Bernotiene et al. [43] to this dataset. We retained only sequences that covered the same 18S rRNA fragment that we sequenced for this study. We combined the GenBank 18S rRNA data with the novel sequences generated for this study to characterize the distributions and host associations of the haplotypes that we identified in North America within a global context. We used the FaBox DNAcollapser tool [44] to reduce the dataset to unique haplotypes prior to analysis and estimated a Bayesian gene tree using BEAST v2.6, implementing the GTR model, a strict molecular clock, constant-size coalescent tree prior, and 10 million generations (with 10% burn-in). We used Leishmania amazonensis isolate (GenBank number JX030088) as the outgroup.”

Results (lines 378-392):

“We used the total 309 sequence dataset (166 from GenBank and 143 novel sequences) to characterize global distributions and host specificities of Trypanosoma haplotypes, while we used a restricted dataset consisting of 147 avian Trypanosoma sequences from North America for species delimitation analyses. The FaBox DNAcollapser tool [44] reduced the global dataset to 84 haplotypes and the North America dataset to 17 haplotypes, which we used to build Bayesian gene trees. The global 18S gene tree revealed that haplotypes with morphological identifications generally clustered together, but were typically not monophyletic with respect to haplotypes without morphological identifications (Fig 2, S1 Fig). Haplotype avium-1 clustered with several other haplotypes that have been attributed to T. avium in a larger clade that contained T. thomasbancrofti. Haplotype SNS-1 fell into a large clade that contained haplotypes of T. anguiformis and T. bennetti, though it was not phylogenetically close to haplotypes that have been morphologically identified as T. everetti from Bernotiene et al. [43]. The global (309 sequence), North America (147 sequence), and GMYC (17 sequence) 18S alignments are available from the figshare data repository associated with this study (DOI: 10.6084/m9.figshare.12869966).

(The main three monophyletic groups (Fig. 2 A) should be matched to the known (morpho)species, as T. bennetti, T. avium, and T. culicavium. I understand, that this is presented as a note, but names in the tree would be more illustrative.)

###

### Author’s response:

###

We have modified Figure 2 following the reviewer’s recommendations. In the revised version we have included a column for morphospecies next to the haplotype phylogeny, so that the reader can easily link each haplotype to its known morphospecies. We do note that we chose not to label the three primary clades as T. bennetti, T. avium, and T. culicavium as the reviewer suggested, as we felt that doing so would be extending the available data beyond what is known. For example, we cannot say with certainty that samples MT276564 and KX179918 are of the morphospecies T. culicavium simply because they are within the same clade. Our understanding of how morphology of avian trypanosomes relates to phylogeny is simply not developed enough to do so.

60-1 Euglenozoa; Kinetoplastea; Trypanosomatida

###

### Author’s response:

###

We have made this change in the revised manuscript (line 62).

65 a possible vector

###

### Author’s response:

###

We have made this change in the revised manuscript (line 67).

101-15 also study of Slapeta et al. (2016) and his analysis should be mentioned in this paragraph.

###

### Author’s response:

###

We have made this change in the revised manuscript (lines 110-115).

“Similarly, Šlapeta et al. [22] found three clades of avian trypanosomes (also referred to as groups A, B, and C) using 18S rRNA data. One clade united T. avium and a novel taxon, T. thomasboncrafti, a second clade consisted of T. corvi and T. culicavium, while a third clade contained mostly unidentified avian Trypanosoma lineages and samples of T. bennetti. These authors found that each avian Trypanosoma clade was more closely related to trypanosomes that infect other groups of vertebrates than they were to each other.”

501-2 Table 2 does not include the mentioned information!!! Tab. 2 includes only previously reported sequences identical with two mentioned haplotypes!!! This information must be presented in a more clear form.

###

### Author’s response:

###

We are thankful that the reviewer noted that this sentence does not match precisely to the information in Table 2. We have now added in the revised manuscript a supplementary table (S1 Table) listing all of the known avian host species for each haplotype. These sentences have been revised to reference this new Supplementary Table (lines 530-534):

“Haplotypes avium-1 and SNS-1 also exhibited a remarkable lack of host specificity: within our newly sampled isolates, we found avium-1 in 17 host species (7 host families), while SNS-1 infected 10 host species (8 host families). After including host records from GenBank, the observed host breadth for each haplotype was even greater – 28 host species (12 host families) for avium-1, and 11 host species (9 host families) for SNS-1 (S1 Table).”

Fig. 2A: MT276439 – 6 host species, but there is no way to find which six species belong to this haplotype, according to the supp. material MT276439=Icterus galbula, but other five species? The supplementary material should be better arranged.

###

### Author’s response:

###

In the revised manuscript we have edited Figure 2 (now Figure 3) and the S2 Appendix to make host species data easy to locate. In what is now Figure 3 we have labeled each haplotype with a number (1 through 17), which correspond to a column in the S2 Appendix. And so if a reader wants to find all host species of the haplotype that MT276439 belongs to, they simply have to note the haplotype number (5) and look for all instances of this haplotype number in the S2 Appendix. 

PLS correct T. culicavum to T. culicavium throughout the text (and Fig. 2A)

###

### Author’s response:

###

We have made this change in the revised manuscript (throughout the manuscript and in revised Figure 3).

---

## [Decision Letter · Decision Letter 1]

18 Sep 2020

Phylogenomics from transcriptomic “bycatch” clarify the origins and diversity of avian trypanosomes in North America

PONE-D-20-16707R1

Dear Dr. Galen,

We’re pleased to inform you that your manuscript has been judged scientifically suitable for publication and will be formally accepted for publication once it meets all outstanding technical requirements.

Kind regards,

Tzen-Yuh Chiang

Academic Editor

PLOS ONE

Additional Editor Comments (optional):

Reviewers' comments:

Reviewer's Responses to Questions

**Comments to the Author**

1. If the authors have adequately addressed your comments raised in a previous round of review and you feel that this manuscript is now acceptable for publication, you may indicate that here to bypass the “Comments to the Author” section, enter your conflict of interest statement in the “Confidential to Editor” section, and submit your "Accept" recommendation.

Reviewer #1: All comments have been addressed

Reviewer #2: All comments have been addressed

2. Is the manuscript technically sound, and do the data support the conclusions?

Reviewer #1: Yes

Reviewer #2: Yes

3. Has the statistical analysis been performed appropriately and rigorously? 

Reviewer #1: Yes

Reviewer #2: N/A

4. Have the authors made all data underlying the findings in their manuscript fully available?

Reviewer #1: Yes

Reviewer #2: Yes

5. Is the manuscript presented in an intelligible fashion and written in standard English?

Reviewer #1: Yes

Reviewer #2: Yes

6. Review Comments to the Author

Reviewer #1: (No Response)

Reviewer #2: The authors have adequately addressed all comments and I have no more comments/notes. Congratulation for the nice study.

7. PLOS authors have the option to publish the peer review history of their article (what does this mean?). If published, this will include your full peer review and any attached files.

Reviewer #1: No

Reviewer #2: No

---

## [Editor Report · Acceptance letter]

29 Sep 2020

PONE-D-20-16707R1 

Phylogenomics from transcriptomic “bycatch” clarify the origins and diversity of avian trypanosomes in North America 

Dear Dr. Galen:

I'm pleased to inform you that your manuscript has been deemed suitable for publication in PLOS ONE. Congratulations! Your manuscript is now with our production department. 

Kind regards, 

on behalf of

Dr. Tzen-Yuh Chiang 

Academic Editor

PLOS ONE